# Torrential rainfall in Valencia, Spain, recorded by personal weather stations preceding and during the 29 October 2024 floods

Nathalie Rombeek<sup>1</sup>, Markus Hrachowitz<sup>1</sup>, and Remko Uijlenhoet<sup>1</sup>

<sup>1</sup>Department of Water Management, Delft University of Technology, the Netherlands

**Correspondence:** Nathalie Rombeek (n.rombeek@tudelft.nl)

Abstract. On 29 October 2024, torrential rainfall locally exceeding 300 mm within less than 24 h, caused devastating floods in the province of Valencia in Spain. In this study we quantify and describe the spatial and temporal structure of the rainfall event on this day using rainfall observations from approximately 225 personal weather stations (PWSs), low-cost commercial devices primarily operated by citizens. The network density of PWSs is ~7 times higher compared to the dedicated rain gauge network operated by the Spanish Meteorological Agency (AEMET) in the province of Valencia, allowing a more detailed analysis of the spatial and temporal rainfall dynamics. In addition, PWS observations are available in near real-time to the public with a temporal resolution of 5 min, whereas the data from AEMET are not available in real time for the public and at a lower publicly available temporal resolution (1 h). Daily rainfall sums recorded by the PWSs showed a high correlation (r = 0.94)and low bias (underestimation of 4%) compared to rainfall reported by AEMET. In the upstream parts of the Magro catchment (1661 km<sup>2</sup>), a first burst of extreme rainfall, reaching up to 180 mm of rainfall in a few hours, started in the morning, leading to the generation of a first flood wave in the upstream parts of the catchment. While the resulting flood wave was propagating downstream through the channel network, a second rainfall peak occurred, which moved downstream along with the flood wave. This spatial and temporal coincidence has likely exacerbated the devastating power of this event. Based on the PWS data, it could have been anticipated that the extreme rainfall already occurring early in the morning would likely result in flooding in the Magro catchment. Areal rainfall maps based on interpolating PWS data indicated catchment average rainfall exceeding 150 mm d<sup>-1</sup> across an area of more than 2500 km<sup>2</sup>. However, the total accumulated rainfall remains uncertain due to interrupted measurements likely caused by power outage and inherent uncertainty associated with interpolating point measurements. For the Rambla de Poyo catchment, the resulting average discharge was around 900 m<sup>3</sup> s<sup>-1</sup>. The estimated return period of the catchment-average rainfall and resulting discharge from this event exhibits large uncertainties, with on average exceeding 10,000 years and 900 years, respectively. This study shows the potential of PWSs for real-time rainfall monitoring and potentially flood early warning systems, by complementing dedicated rain gauge networks in order to reduce the uncertainty from areal rainfall estimates and to localize potential flooding more accurately.

## 1 Introduction

On 29 October 2024, the eastern part of the Province of Valencia, Spain, was struck by torrential rainfall, locally exceeding 300 mm within 24 h (Faranda et al., 2024). This was caused by an isolated high-altitude low-pressure weather system that was

separated from the jet-stream (a so-called cut-off low) (AEMET, 2024a; Faranda et al., 2024). This phenomenon is locally known as DANA, a type of weather system that has struck the Valencia region already several times in the past, notably in 1982, 1987 and 2007 (Egozcue and Ramis, 2001; Peñarrocha et al., 2002; Pastor et al., 2010; Ferreira, 2021). The extreme rainfall amounts combined with the mountainous, partially urbanized and possibly rapidly responding catchments, exacerbated by challenges in early warning and response, resulted in at least 221 fatalities and destruction of infrastructure (Llasat, 2024).

Extreme rainfall can trigger floods in both mountainous areas and lowland catchments (Miglietta and Regano, 2008; Gaume et al., 2009; Brauer et al., 2011; Gochis et al., 2015; Marchi et al., 2010). Because of the short response times of urbanized mountainous areas (as short as 20 min for steeply sloping catchments of around 10 km², Morin et al. (2001); Berne et al. (2004)), flood early warning systems heavily rely on short-term meteorological forecasts (nowcasts). Accurate and timely available rainfall observations with a high spatial and temporal resolution are essential for flood early warning systems (Bárdossy et al., 2021; Bárdossy and Anwar, 2023).

Weather radars and rain gauges are instruments commonly used by national meteorological and hydrological services (NMHSs) in Europe to record rainfall. Rain gauge networks from NMHSs provide accurate measurements, however, the network density is typically insufficient for providing robust areal rainfall estimates (e.g. on average 1 gauge per 600 km² in the province of Valencia) and are not always available in near real time. On the other hand, weather radars estimate rainfall indirectly and their rainfall retrievals are prone to several sources of error, resulting in substantial uncertainty and bias in the rainfall estimates (Krajewski et al., 2010; Villarini and Krajewski, 2010). Moreover, rainfall estimation in mountainous areas is particularly challenging for weather radars, as mountain ranges can (partially) block radar beams (Pellarin et al., 2002; Germann et al., 2006). Radar-rain gauge merging typically improves the quantitative precipitation estimates, nevertheless, the sparse network of dedicated rain gauges can not fully compensate for sources of errors in radar estimates (Goudenhoofdt and Delobbe, 2009; Ochoa-Rodriguez et al., 2019; Overeem et al., 2024).

Rain gauges from personal weather stations (PWSs), which are low-cost commercial instruments primarily owned and operated by citizens, can be used for observing rainfall in real time. These PWSs are typically not installed and maintained according to international guidelines. Once connected to an online platform, rainfall data is automatically uploaded to online platforms such as Netatmo (https://weathermap.netatmo.com/), Weather Observations Website (WOW; https://wow.metoffice. gov.uk/) and wunderground (https://www.wunderground.com/wundermap) and data is accessible from the connected online platform in near real time with a high temporal resolution (5 min). Over the last decade, the abundance of PWSs has increased markedly across Europe. Notably, tens of thousands of PWSs from the Netatmo brand alone are nowadays in operation, contributing to a widespread network of PWSs (Overeem et al., 2024; Rombeek et al., 2025). While network densities can be greater than those of dedicated rain gauge networks operated by NMHSs (e.g. roughly 7 times more PWSs in the province of Valencia), PWSs are prone to several sources of error, related to setup, maintenance, data processing and instrumental errors (De Vos et al., 2017). Despite this, previous studies have demonstrated that, with proper quality control, PWSs can potentially be used for providing and enhancing real-time rainfall estimates (De Vos et al., 2017, 2019; Bárdossy et al., 2021; Graf et al., 2021; Overeem et al., 2024; Nielsen et al., 2024; Rombeek et al., 2025). For example, De Vos et al. (2017) used an experimental setup to show that under ideal conditions (i.e. installed and maintained according to World Meteorological

Organization standards), three PWSs collocated with a floating-type gauge from the Royal Netherlands Meteorological Institute (KNMI) recorded rainfall with high accuracy. More recently, Rombeek et al. (2025) performed a systematic long-term analysis of PWS rainfall observations by comparing them against KNMI's professional rain gauge network (float-type), and reported an overall high performance of the PWSs, especially for longer accumulation intervals. Furthermore, data from PWSs are already employed by meteorological agencies, such as the local meteorological association of the Region of Valencia (AVAMET).

Here, we investigate the spatial and temporal structure of the torrential rainfall events that resulted in the devastating floods in the Valencia region, by making use of rainfall observations from personal weather stations. We quantify and report on the extreme rainfall that triggered the floods. We show the potential and limitations of using personal weather stations for real-time monitoring of heavy rainfall.

#### 70 2 Study area

75

This study was carried out over part of the Valencia Province, within the Spanish Mediterranean coastal zone (Fig. 1a). This region is characterized by a narrow coastal strip of a few km and a complex topography with steep slopes and mountain ranges exceeding 1400 m (Fig. 1b). The coastal area has become heavily urbanized over the last decades, increasing flood risks (Camarasa-Belmonte and Soriano-García, 2012). Other urban settlements can be found in the valleys, typically close to river channels.

The Province of Valencia is semi-arid, with yearly rainfall between 1991 and 2020 ranging from 300 to 600 mm yr<sup>-1</sup> (Chazarra-Bernabé et al., 2024). More than 50% typically falls on the 10 days with the highest rainfall accumulations (González Hidalgo et al., 2003). Orographic convection plays an important role in the formation of heavy rainfall in the Valencia region (Pastor et al., 2010). The average reference evaporation between 1996 and 2020 was between 1100 and 1200 mm yr<sup>-1</sup> (Chazarra-Bernabé et al., 2024). Previous research indicated that this region is prone to torrential rainfall, in particular during autumn, with return periods of less than 5 years for daily rainfall of 100 mm d<sup>-1</sup> and 75 years for 200 mm d<sup>-1</sup> (Romero et al., 1998). More recently, Lazoglou and Anagnostopoulou (2017) estimated that daily rainfall totals exceeding 300 mm d<sup>-1</sup> along the Spanish Mediterranean coast are associated with return periods in the range of 150 to 300 years.

The downstream areas of the Turia and Júcar catchments were most affected by the torrential rainfall on 29 October 2024 that resulted in devastating floods. These rivers drain south of the city of Valencia into the Mediterranean Sea. The Rambla de Poyo catchment, located between these rivers, also experienced severe flooding on 29 October. For that reason, this assessment focuses on the downstream area of the Turia river, on a tributary of the Júcar river, namely the Magro river, and the Rambla de Poyo (Fig. 1). Daily average discharge measurements for the streamflow gauges in Fig. 1b on 28, 29 and 30 October 2024 were obtained from the Júcar Hydrographic Confederation. The upstream areas and discharge are shown in Table 1, indicating that the measured average discharge values on 29 and 30 October reached up to seven times the yearly average peak, meanwhile the antecedent conditions were below average. The average discharge on 30 October represents the highest recorded value for gauge ID1 throughout the measurement period. The Poyo catchment experienced runoff per unit area on the 29th and 30th that

Figure 1. Study area, including catchments that were affected by the torrential rain on 29 October 2024. a) Yellow area in the overview map indicates the Province of Valencia. b) Dashed catchment boundaries correspond to the upstream area of the streamflow gauging stations (ID1-ID4). Yellow catchment boundaries are used for calculating catchment average rainfall. Catchment delineations were obtained from Lehner and Grill (2013) and Do Nascimento et al. (2024). c) Locations of the employed personal weather stations (PWSs) and the rain gauges from the Spanish Meteorological Agency (AEMET). Capital letters (A-F) in red next to six PWSs correspond to the time series that are shown in Section 4.2 (Fig. 5). Black hatched area indicates the municipality of Valencia. Digital elevation model (DEM) obtained from European Space Agency (2024), river network from Copernicus Land Monitoring Service (2019) and base map from ©OpenStreetMap (www.openstreetmap.org). Major towns in the area are indicated with black stars (Valencia, Utiel, Chiva, Montroi and Algemesí).

was up to 100 times higher than measured for the other catchments (Table 1). Unfortunately, no detailed discharge time series during and after the event were available to us.

In the past, flash floods occurred frequently in these catchments (Ruiz et al., 2014; Camarasa-Belmonte, 2016). In the Rambla de Poyo catchment, characterized by an ephemeral stream with impermeable lithology and an average slope of 17%, nearly 40 flash flood events were recorded in less than 20 years (Camarasa-Belmonte, 2016). The Turia river, which passed through the downtown Valencia, was struck by a major flood event in October 1957. Afterwards, this river was diverted south of the city to limit the fatalities and damage in the future (Portugués Mollà, 2024).

**Table 1.** Area and mean discharge per unit area of the gauged catchments indicated in Fig. 1b. Mean discharge and mean yearly peak discharge for gauges ID1, ID2 and ID3 are calculated from mean daily streamflow data in Do Nascimento et al. (2024). The Júcar Hydrographic Confederation provided mean daily discharge observations on 28, 29 and 30 October 2024. "NA" means that data was not available.

|                  |                         |             | Mean daily discharge per unit area [mm d <sup>-1</sup> ] |                  |            |            |            |
|------------------|-------------------------|-------------|----------------------------------------------------------|------------------|------------|------------|------------|
| Streamflow gauge | Area (km <sup>2</sup> ) | Time period | Mean                                                     | Mean yearly peak | 28-10-2024 | 29-10-2024 | 30-10-2024 |
| ID1 (Júcar)      | 21331                   | 1947 - 2018 | 0.12                                                     | 0.85             | 0.03       | 1.33       | 5.66       |
| ID2 (Magro)      | 706                     | 1916 - 2018 | 0.07                                                     | 1.81             | 0.02       | 3.34       | NA         |
| ID3 (Turia)      | 6123                    | 1916 - 2018 | 0.15                                                     | 0.76             | 0.11       | 1.16       | NA         |
| ID4 (Poyo)       | 182                     | NA          | NA                                                       | NA               | 0          | 147        | 433        |

#### 3 Data and methods

#### 3.1 Personal weather stations

For the analysis, we used data from PWSs of the Netatmo brand operational and available around the province of Valencia (Fig. 1). This weather station measures by default temperature, pressure and humidity and can be extended by a rain gauge and anemometer to measure rain and wind, respectively. We only selected PWSs that were equipped with a rain gauge. This data, available in near real-time, is archived at 5 min temporal resolution and is freely accessible using an application programming interface (API) (Netatmo, 2024, last access: 21 March 2025). In total 245 PWSs with a rain gauge extension are located in the study area and used in this analysis as shown in Fig. 1, which represents 37% of the total PWSs in this area. The rain gauge of Netatmo PWSs is a tipping bucket, with a collecting funnel of 13 cm diameter (133 cm<sup>2</sup> orifice) and a nominal tipping volume of 0.101 mm according to the manufacturer (Netatmo, 2025). According to the manufacturer, the nominal accuracy is 1 mm  $h^{-1}$  for a measurement range of 0.2 to 150 mm  $h^{-1}$  (Netatmo, 2025). These nominal values may not necessarily hold in practice. For example, extensive calibration of another type of low-cost rain gauge (i.e. Davis) in a laboratory revealed deviations in both tipping volume and measurement accuracy from the nominal values (Krüger et al., 2024). It is expected that similar discrepancies apply to the Netatmo rain gauges, as these also use tipping bucket mechanisms. In addition, the tipping bucket volume of tipping bucket rain gauges is not constant, but depends non-linearly on rainfall intensity (Marsalek, 1981; Niemczynowicz, 1986; Humphrey et al., 1997). In this work, to account for potential deviations from the nominal value, a fixed mean bias correction factor of 1.24 is applied to the downloaded data, as this was found to yield satisfactory results in previous studies using the same type of PWS (De Vos et al., 2019; Rombeek et al., 2025).

Approximately every 5 min data from the rain gauge module is wirelessly transmitted to the indoor module. The indoor module transfers the data to an online platform using Wi-Fi. The Netatmo software assigns the measured rainfall to the next full five-minute interval. When the connection between these modules is temporarily interrupted, any rainfall measured during the interruption is accumulated and assigned to the timestamp when the connection is reestablished. If the indoor module does not have any power supply, data is lost and consequently not reported.

**Figure 2.** Histogram of nearest neighbour distances of a) PWSs and b) AEMET stations. Vertical red dashed line indicates the mean distance, vertical black line the median, the left and right whiskers indicate 1.5 times the inter-quartile range from the lower (left) and upper (right) box, boxes the inter-percentile range (25th–75th) and small circles the outliers.

From the total set of PWSs in the study area, we discarded 20 PWSs ( $\sim$ 8%), of which 15 PWSs exhibited irregular data transmission (e.g. no data transferred for 11 consecutive hours) for at least 40% of the time on 29 October, and five PWSs were likely reporting incorrect zero rainfall values.

#### 3.2 Reference data

The Spanish Meteorological Agency (AEMET) operates 40 automatic rain gauges with a 5-min temporal resolution in the study area as shown in Fig. 1b,c. Data up to seven days prior can be accessed from the website using an API (https://opendata.aemet.es/, accessed on: 4 November 2024). The rainfall data on the 29 October 2024 in the Valencia region was downloaded and used for comparison. Total daily sums, as well as 6-hourly precipitation totals are provided in local time (at 06:00, 12:00, 18:00 and 00:00 local time). As there are some gaps in the data, we assume that the data provided on the website did not yet undergo any quality control. We discarded one rain gauge in the province of Castelló that did not have any observations available on 29 October.

The distance to the nearest neighbouring rain gauge is on average smaller for the PWSs than for the AEMET stations (median of 1.7 and 18.3 km, respectively) (Fig. 2). While the number of PWSs is larger than the number of AEMET stations in the area shown in Fig. 1c, the PWSs are more concentrated around urban areas, which are particularly along the coast, resulting in a spatially inhomogeneous network.

## 3.3 Catchment average rainfall

The hydrological response depends on the areal rainfall. For that reason, the rainfall averages of several catchments were estimated based on catchment delineations from Hydrosheds (Lehner and Grill, 2013). Rainfall maps were obtained by interpolating rain gauge measurements from AEMET and PWSs using ordinary kriging (OK) with an isotropic spherical variogram model (Delhomme, 1978). The range was based on fitting a new relation between aggregation time and correlation distance, derived from the relations found by Lebel et al. (1987) and Berne et al. (2004) for intense Mediterranean rainfall events (see supporting information, Fig. B1). These relations were both based on a spherical variogram model. This resulted in the following power-law relation between decorrelation distance and aggregation time:

$$d = 5.32\Delta t^{0.36} \tag{1}$$

in which d is the decorrelation distance in km and  $\Delta t$  the time interval in min. Similarly to Van de Beek et al. (2011), the nugget was assumed to be zero. The sill was estimated as the variance of the rainfall data on which the variogram was based.

## 3.4 Return periods

To estimate the return periods of the rainfall and measured discharges, extreme value statistics were used. For the rain gauges with the highest rainfall recordings, no sufficiently long timeseries were available. Instead, timeseries of two other AEMET rain gauges in the Province of Valencia were used and obtained from Klein Tank et al. (2002). The catchment average rainfall timeseries was estimated for catchment ID2 in Fig. 1b. These timeseries were derived from the gridded observational precipitation dataset E-OBS (Cornes et al., 2018). The catchment-average rainfall dataset is included in the EStreams dataset (Do Nascimento et al., 2024). For three catchments (ID1-ID3 in Fig. 1b and Table 1) discharge timeseries from the EStreams dataset were used (Do Nascimento et al., 2024). Generalized extreme value (GEV) distributions were fitted to the annual maximum rainfall and discharge values (Jenkinson, 1955) including a shape parameter (Weibull-type). The distribution parameters (μlocation, α scale and κ shape) were estimated through the log likelihood estimation using the Nelder-Mead optimization algorithm (Overeem et al., 2010). To quantify uncertainty in the fitted GEV parameters, bootstrapping by sampling with replacement was applied using 1000 random samples (Efron and Tibshirani, 1994).

#### 4 Results


#### 4.1 Quality assessment

To quantify the performance of the rainfall observations from PWSs in the area, they were evaluated against the daily rain gauge data from individual AEMET gauges, using Pearson correlation coefficient (r), the relative bias and the slope of the fitted linear regression line (a). Only the closest PWSs within 10 km from an AEMET station were selected to limit the influence of spatial rainfall variability, in line with Rombeek et al. (2025). This resulted in 28 PWS-AEMET pairs. Only AEMET stations with complete daily data, without 6 h data gaps, were used for the analysis. To allow for a fair comparison, we only selected PWSs

Figure 3. Scatter plot of 24-h rainfall rate between PWS-AEMET pairs with a maximum inter-gauge distance of 10 km, where r indicates the correlation coefficient, bias indicates the average relative bias over all pairs and  $R_{\text{PWS}} = aR_{\text{AEMET}}$  represents the linear regression line through the origin in red, with a indicating the slope. The red shaded area represents the ranges of slopes and  $\pm$  represents the standard deviation derived from bootstrapping.

that did not experience a power failure on 29 October and for that reason were unlikely to have missing data. This resulted in 24 PWS-AEMET pairs. As shown in Fig. 3, a high correlation (r = 0.94) between the PWSs and AEMET stations is observed. After applying a default bias correction factor of 1.24, the relative bias is 4%, indicating that on average the PWSs slightly underestimate the rainfall. The average of the slopes of the fitted lines through the origin is 0.93, indicating a proportional underestimation. Bootstrapping by leaving-one-out approach reveals uncertainty in the fitted line, with slopes ranging between 0.83 and 1.05. The bias and correlation coefficient show little to no spread, indicating overall consistency.

Rainfall observations from 29 October from two selected pairs of PWSs that are located within 1 km from each other were compared in more detail. To assess the spatial variability and uncertainty of the measurements, the correlation coefficient was calculated. From Fig. 4a it is observed that both station pairs show a good correlation (r = 0.93 and r = 0.91), indicating consistency in the observations. When accumulated over 1 h and only including 5 min where both stations had data available and no previous time step was missing (to avoid unrealistically large accumulations, i.e. high influxes), the correlations become 0.99 and 1.0 for the stations in Algemesí and Requena (see supporting information, Fig. A1), respectively (locations E and B in Fig.1c). The double mass plots indicate that there is a systematic difference between the two stations in the town Requena and

Figure 4. a) Scatter plot of 5-min rainfall between two pairs of PWSs (in Requena and Algemesí, indicated with B and E in Fig. 1c), with an inter-gauge distance of less than 1 km. The correlation coefficient (r) quantifies the relation between the stations. b) Cumulative rainfall between two pairs. Only intervals where both stations contain measurements are included. Legend indicates in which town the PWSs were located and the inter-gauge distance.

in Algemesí. In the town Requena, PWS1 recorded 19 mm<sup>-1</sup> more than PWS2, while in the town Algemesí PWS1 recorded approximately 50 mm more than PWS2, namely 186 mm (Fig. 4b).

#### 4.2 Rainfall time series




Intense rainfall exceeding  $400 \text{ mm d}^{-1}$  was reported by AEMET for a minimum of one rain gauge in the Province of Valencia. Similarly, PWSs recorded high rainfall intensities and accumulations. Figure 5 illustrates the variations of the rainfall intensities and accumulations over local time and in space for six selected PWSs which recorded more than  $100 \text{ mm d}^{-1}$ . The locations of the stations are indicated in Fig. 1c. Within the Magro river catchment, most rainfall occurred in the 14 hour period between 06:00 and 20:00. The timing of the peaks and the rainfall depth varied within this region.

In the headwaters of the Magro river two rainfall peaks were recorded by the stations A and B (distance between gauges ~12 km; elevations 740 m and 705 m, for A and B, respectively). The first peak occurred in the early morning between 06:00 and 08:30 and the second peak around 11:30 (Fig. 5A,B). At location A, around 15:45, data transfer was frequently interrupted, likely due to temporary power failure, as the indoor module also failed to provide data for two or more consecutive time intervals. These temporary connection issues suggest a potential underestimation of the actual rainfall sum as the AEMET station in Utiel, at a distance of ~4 km, reported 13.2 mm between 18:00-00:00. Similarly, for PWS B, between 15:05 and 15:50, data transfer was interrupted likely due to temporary power outage. Around 16:15 the connection between the indoor

**Figure 5.** Time series of the 5-min rainfall observations from six selected PWSs that recorded more than  $100 \text{ mm d}^{-1}$  on 29 October 2024 in local time. Letters in the figure correspond with the locations (indicated with red capital letters) in Fig. 1c. Left y-axis shows the intensity in mm h<sup>-1</sup>, right y-axis the cumulative rainfall in mm. The red dashed line indicates when measurements are (temporarily) interrupted and total rainfall sums become uncertain. In the black dotted box the outline of the Magro catchment is shown, including the location of the PWS displayed in the panel (red dot) and the locations of the other five PWSs (black dots).

and rain gauge module was temporarily interrupted, accumulating the data to the time when a connection is established again, resulting in a high influx at 16:50. More downstream, at locations C and D (distance between gauges  $\sim$ 14 km; elevations 131 m and 56 m, for C and D, respectively), the onset of rainfall was earlier, however, the first peak started approximately 45 min later than at stations A and B, around 06:45. Around 10:00, this first burst of rainfall ended. At location C, the highest rainfall intensities, frequently exceeding 100 mm h<sup>-1</sup>, were recorded between 07:35 and 08:55. The second peak occurred a few hours later than at stations A and B, around 14:30 (Fig. 5C,D). Both stations abruptly stop recording any rainfall. Station C, only reports 0 mm values after 16:15 likely due to malfunctioning of the gauge (e.g. tilted) and stops recording any data on 30 October at 07:00. The rain gauge module of PWS D, which is located  $\sim$ 300 m from the Magro river, lost connection with the indoor module for  $\sim$ 20 min after 15:15. During this period, pressure measurements remained available, suggesting that it might be expected that the accumulated rainfall would be assigned to the time when the connection was restored. However, rainfall measurement remained 0 mm until 17:20. It is unknown whether this behavior was due a malfunction of the rain gauge. After 17:20, no data is available likely due to a power failure. More downstream, at location E (elevation 25 m), most of the precipitation, around 150 mm, occurred between 16:00 and 22:00 (Fig. 5E). Within this time period, the connection between


the indoor and rain gauge module was temporarily interrupted, likely resulting in a high influx at 21:30. Station F (elevation 39 m) is located close to the Júcar river, upstream from the confluence with the Magro river. Between 05:00 and 10:00, this PWS recorded most rainfall, nearly 185 mm was recorded in 5 hours. After 11:10 an additional 45 mm of rain was reported, until around 14:30.

Within a 24-h period, 5-min rainfall intensities exceeding 75 or 100 mm h<sup>-1</sup> were frequently recorded. At four PWSs located along the Magro river (Fig. 1c: PWS A, B, C and E), rainfall accumulations exceeding 200 mm d<sup>-1</sup> and reaching up to 300 mm d<sup>-1</sup> were recorded. For PWS A, B, C and D, the rainfall recordings become uncertain between 15:10 and 16:15, suggesting significant underestimation of the total rainfall on that day. The magnitudes reported therefore need to be understood as lower limits, i.e. it has rained at least that much, but probably considerably more.

Reconstructing the dynamics of the rainfall events on 29 October, measurements from the PWS network suggest that rain started early in the morning in the south, around location F, and moved north-west, in the upstream direction of the Magro catchment. This first burst of extreme rainfall led to accumulated rainfall ranging from 60 and 160 mm within approximately three hours over PWS A, B, C and D, with PWS C reporting more than 100 mm in one hour. These rainfall intensities likely exceeded soil infiltration capacities, generating Hortonian overland flow (Horton, 1933) across the hilly catchment and potentially triggering a first flood wave. The second peak of rainfall was first recorded upstream, at location A and B, and then moved downstream. This suggests that the east-moving trajectory of the second rainfall peak coincided with the downstream propagation of the flood wave triggered by the first peak. It is thus plausible to assume that the superposition of the second rainfall peak with the first flood wave considerably amplified the downstream flood magnitude

## 4.3 Catchment average rainfall






For hydrological applications, the required spatial and temporal resolution of rainfall measurements depends on the hydrological response of a catchment. In order to give an indication about the response times of the affected catchments, the power law relation from Berne et al. (2004) ( $t = 0.75S^{0.3}$ ), with t in minutes and S the catchment area in ha) was used, which relates the response time in urban mountainous catchments to the surface area of the catchment. According to this power law, response times between 1 and 2 h can be expected for the smallest (176.7 km<sup>2</sup>) and largest catchment (2104 km<sup>2</sup>) in Fig. 1. For that reason, hourly catchment averages are calculated. For hydrological applications, such as flood forecasting, the minimum required temporal resolution should be around a factor 4 smaller than the response time (Schilling, 1991; Berne et al., 2004). For the catchments considered here, this implies a required temporal resolution between 15 and 30 min. As PWSs have a temporal resolution of 5 min, this should be sufficient for real-time hydrological applications.

Selected time series are limited to the catchments with the highest accumulated rainfall within 24 h and which were severely flooded (Fig. 6). The rainfall distributions in time over these catchments show that most of the rainfall occurred in a period less than 24 h (Fig. 6a and Fig. 7). The highest catchment-average hourly peaks (30 to 35 mm h<sup>-1</sup>) occurred in the largest catchment (3). While for catchments 1, 2 and 3 most rainfall occurred at 08:00, this was at 06:00 for catchment 4. This indicates that rainfall started in the south in the morning and moved north. Within 24 h, catchments 2, 3 and 4 reached average rainfall sums between 150 and 200 mm d<sup>-1</sup> according to the interpolated PWS data. In contrast, the average rainfall for catchment 1

was significantly lower, around 70 mm. Most rain occurred more inland (World Meteorological Organization, 2024), which is the southwestern part of catchment 1, while the other part of the catchment received significantly less rain. Especially in the east only daily sums between 5 and 30 mm  $d^{-1}$  were recorded. This uneven distribution of rainfall over the catchment contributed to the lower average rainfall in catchment 1.

The average interpolation uncertainty associated with kriging is low (less than 1 mm h<sup>-1</sup>, Fig. 6a), however, this does not account for the uncertainty arising from poor setup and maintenance of the PWSs and instrument malfunctioning due to power outage or damage resulting from flooding. For certain PWSs recordings became uncertain later during the day, although these have not been excluded here (Sect. 4.2). This suggests that the estimated catchment average rainfall was likely underestimated. Interpolating daily rainfall based on PWSs and AEMET stations results in somewhat different catchment-average rainfall sums (Fig. 6b,c). While Fig. 3 showed that daily rainfall sums from the PWSs are highly correlated with the AEMET stations and have an average underestimation of only 4%, the differences in network density and distribution affect the interpolation. For example, in catchment 1, the network density is around 10 times higher for the PWSs (1 PWS per ~32 km<sup>2</sup>) than for the AEMET stations (1 gauge per 320 km<sup>2</sup>). In addition, the records of one AEMET station, located in Túris, largely influenced the average rainfall in catchment 1. This station reported more than 400 mm within 24 h, with most rain (around 280 mm) between 18:00-00:00, while PWSs within 10 km did not record this. The PWS south of Túris, in the town of Montroi (at 5.42 km distance), recorded around 300 mm until 18:00. However, the total rainfall of this PWS is uncertain (see Fig. 5C). The PWSs north of Túris (gauge distances between 5.35 and 8.71 km) recorded 24 h rainfall sums between 25 and 153 mm. One of these gauges was manually calibrated by its owner with a factor of ~6 lower than the nominal tipping volume calibrated by the manufacturer. Changing this tipping volume to the nominal value of 0.101 mm results in a rainfall sum of 125 mm.

Figure 6d shows the areal rainfall maps based on combing both AEMET and PWS rain gauge network. This results in a larger network density and consequently to a lower uncertainty associated with kriging and potentially providing more accurate catchment-average rainfall estimates compared to Fig. 6b,c. Combining these rain gauge networks results in lower catchment-averaged rainfall estimates compared to AEMET data alone for catchments 1), 2) and 3) (Fig. 6b,d). The largest reduction occurs in catchment 1), with a decrease of approximately 60 mm, resulting in a catchment-average rainfall of 79 mm. On the other hand, for catchment 4) a significantly higher catchment-averaged rainfall is estimated, which is more than 40 mm, namely 164 mm. These estimated values are closer to the areal rainfall map based on PWSs, with a difference of approximately 10 mm (Fig. 6c,d). These results indicate that including rainfall observations from PWSs has a large effect on the estimated areal rainfall maps and likely provides more accurate areal rainfall maps at the locations where the dedicated rain gauge network is sparser. In this analysis, the uncertainty in rainfall observations, which is expected to be greater for PWSs than that of AEMET, is not taken into account.

## 4.4 Spatio-temporal evolution






The spatio-temporal evolution of the storm in a geographical context is shown in Fig. 7. These maps show that during the first 6 hours (00:00 - 06:00, local time) the rainfall was concentrated over catchments located in the south of the study area. Over the next 6 hours, the storm's spatial extent increased, predominantly growing in the north-westerly direction. Rainfall intensities

increased substantially, with the Magro catchment and adjacent catchments receiving the most rainfall. Between 12:00 and 18:00, the spatial extent of the storm decreases, with highest rainfall intensities still occurring in the central part of the study area. Based on the available rainfall data, the storm's spatial extent and rainfall intensities appear to decrease further during the last 6 hours, with most rainfall concentrated in the catchments in the central and northern parts of the study area. Note that the rainfall observations during the end of this period are uncertain due to for example (temporary) power failure (see Fig.5).

### 4.5 Return periods




To put this event into a wider context, extreme precipitation statistics were derived from two AEMET-operated rain gauges near the city of Valencia, located approximately 11 km apart. Although these gauges did not record high 24-h rainfall sums on 29 October, their relatively long timeseries provide a rough estimate of the return period for similar rainfall events in the area, as they represent the same climate. The estimated return period for 300 mm d<sup>-1</sup> is around 2,000 years at the Valencia gauge and 8,600 years at the Valencia airport gauge (Fig. 8a). These estimated return periods vary significantly between the two gauges, with more uncertainty for the Valencia airport gauge, which has a shorter record (58 years of data) compared to the Valencia gauge (87 years of data).

Especially for intense events, return periods over larger areas are by definition longer compared to rainfall of the same amount at one location. To illustrate this, 74 years of gridded 24-h precipitation data from E-OBS for catchment ID2 in Fig. 1b was used. Catchment-average rainfall on 29 October over this catchment resulted in daily average rainfall of 194 and 186 mm based on the AEMET or PWS gauges, respectively. This is approximately 3 times higher than the highest observed catchment average rainfall within the 74 year period. The extent of this event is also reflected in Fig. 8b, suggesting that the return period for this event at the catchment scale may be far more than 10,000 years.

The extreme rainfall triggered dramatic floods in the area. To put these flood events in context, we estimated the return periods corresponding to the measured discharges by the local waterboard (Confederación Hidrográfica del Júcar) in the three catchments that were heavily flooded (ID1-ID3, Fig.1b). Based on the fitted GEV distributions, the estimated return period for the average daily discharge observed on 29 October is approximately 5 years across all three catchments (Fig. 9). For both the Magro and Turia no data is available for the subsequent days. For the Júcar catchment, the highest daily average discharge triggered from this rain event was observed on 30 October, which corresponds to an estimated return period of approximately 900 years. Due to a limited size of archived discharge data, the GEV model introduces uncertainty. Rarer events are less represented and thus result in more uncertainty. The 95% confidence interval for the estimated return period of the discharge observed on October 30th ranges from approximately 100 to 6,000 years.

#### 305 5 Discussion






## 5.1 Data uncertainty

Multiple PWSs that were located in the area where most rainfall occurred abruptly stopped measuring. For that reason, the results shown in this study indicate the minimum amount of rainfall that occurred. Reasons can be power failure, as suggested by missing pressure measurements, debris or sediments blocking the orifice of the bucket, tilted devices or flooded devices as a consequence of the intense rainfall and resulting floods. While the actual rainfall is likely to have been higher, the rainfall measurements from the PWSs provide insights in the trajectory and timing of rainfall in space.

The rain gauges from AEMET are also prone to these complications. This could explain missing time steps in the data obtained from the website. In addition, the timing of the recordings may have been influenced by power outage or clogging of the gauge. Notably, this was observed for the station in Túris, which did not have any observation available between 12:00 and 18:00, while between 18:00 and 00:00 more than 270 mm was reported.

Rain gauges from PWSs can be manually calibrated by their owners. Around 17% of the PWSs used in this study were manually calibrated, with 95% of the calibrated tipping volumes falling between 0.09 and 0.15 mm. The accuracy of these calibrations remains unknown. To assess the impact of the calibration, the tipping volume was adjusted to the calibrated tipping volume by the manufacturer (0.101 mm). The effect of this manual calibration on catchment-average rainfall was minor, resulting in an average change of 3%.

# 5.2 Rain gauge network

The distribution of the PWS network allows a more detailed analysis of the rainfall dynamics over the Magro river catchment. In this catchment area (1661 km², catchment 3 in Fig. 6) 8 PWSs are located and only one AEMET station. While the network density of the PWSs in this catchment is fairly low, namely one PWS per 208 km², they are fairly homogeneously distributed over the catchment, providing insight in the rainfall dynamics and trajectory of the storm, which is not possible with only one AEMET station in this catchment. However, for the Turia basin, most urban areas, and thus PWSs, are located in the northeast part of the basin near the coast (e.g. catchment 1 in Fig 6), while most rainfall likely occurred in the southwest part of this catchment (World Meteorological Organization, 2024). This limits the analysis and consequently getting insight in the rainfall dynamics in that area.

Interpolating point measurements inherently leads to uncertainties in areal average rainfall estimates. The degree of uncertainty is strongly dependent on density and homogeneity of the rain gauge network used (Hrachowitz and Weiler, 2011; Wang et al., 2015). The inhomogeneous PWS network induces for that reason uncertainty in the interpolation. On the other hand, the sparser network density of the AEMET stations leads to more kriging variance and consequently more uncertainty in the areal rainfall estimates. For that reason, the catchment average rainfall map based on the AEMET stations in Fig. 6b should not be interpreted as ground truth in this study. Combing both rainfall observations from AEMET and PWSs reduces the kriging variance. Ordinary kriging does not take varying measurement uncertainty into account, potentially introducing errors or biases in

the areal rainfall maps. Alternatively, a more sophisticated method, such as kriging for uncertain data, which handles uncertain data of each rain gauge individually, could improve areal rainfall estimates (Cecinati et al., 2018).

The catchment areas were based on hydrosheds delineations. This resulted in relatively large catchment areas, which limits this analysis in providing insight in the more local variability. Selecting smaller catchment areas would increase the uncertainty in the interpolated rainfall estimates based on point measurements, particularly for areas with no or only a few rain gauges. Rainfall estimates from weather radars can potentially mitigate this gap. From a hydrological perspective, using relatively large catchment areas is not a limitation either, especially given the lack of high temporal resolution discharge observations. Additionally, smaller catchment areas would not necessarily match the upstream areas of streamflow gauges.

## 345 5.3 Return periods






The areal rainfall maps showed that a tributary of the Júcar river, namely in the Magro river, was struck by extreme rainfall, with an estimated return period by far exceeding 10,000 years. Based on this, it is likely that the discharge from the Magro played a large role in contributing to the discharge observed in the Júcar. While no discharge measurements were available on 30 October for streamflow gauge ID2, the likely dominant contribution of the Magro river to the extraordinary discharge of the Júcar (with an estimated return period of approximately 900 years) and the intense rainfall over this catchment suggests that the Magro's discharge return period was also extreme.

The Poyo catchment is characterized by an ephemeral stream (Camarasa-Belmonte, 2016). No historic discharge timeseries longer than three months were available for the streamflow gauge in this catchment (ID4). Consequently, mean and mean yearly peak discharge could not be derived for this gauge. To put the 29 October 2024 discharge levels at ID4 into perspective, we reviewed literature that studied the same catchment. Camarasa-Belmonte (2016) collected discharge data from 38 flash flood events that occurred in the Poyo catchment between 1989 and 2007. They found that the mean and maximum specific peak flow from these 38 events were 16 and 246 mm d<sup>-1</sup>, respectively. In comparison, the recorded discharge on 29 and 30 October 2024 were 147 and 433 mm d<sup>-1</sup> (1), approximately 9 and 27 times higher than the mean peak flow from the 38 flash flood events recorded between 1989 and 2007. The average discharge on 30 October 2024 was significantly higher (1.75 times) than the maximum specific peak flow recorded between 1989 and 2007. Due to the short duration of the historical record used in Camarasa-Belmonte (2016) (18-years), no definitive conclusions can be drawn regarding the return period of this event.

During this particular flood event, a significant quantity of urban drifters and floating debris were observed, impacting the operational efficiency of hydraulic infrastructure and subsequently influencing the propagation of the flood waves across the catchments, especially in urban areas. Consequently, this led to increased flooding and damage, underscoring the importance of accounting for such factors in future extreme events.

## 5.4 Opportunities for PWSs

PWSs from the Netatmo brand are automatically connected to their own platform. Alternatively, citizens can manually register their device to other weather platforms such as the Weather Observations Website (WOW) and Wunderground. These platforms collect meteorological data from all types of PWSs in near real-time. However, data is not necessarily freely ac-

cessible. Nevertheless, national or local meteorological and hydrological services might use these platforms. For example, the local meteorological association in Valencia (AVAMET) collects rainfall measurements from PWSs. These are different brands of PWSs (i.e. tipping bucket rain gauges from the brands Davis and Sainlogic) than used in this study. This network has a higher density in the Valencia region. In addition, higher 24 h rainfall accumulations were recorded by these stations (e.g. more than 600 mm in Túris). These stations are, similarly to the PWSs used in this study, not necessarily installed and maintained following the world meteorological organization (WMO) guidelines, because some of the metadata indicate that these rain gauges were installed on roofs (e.g. station in Túris, see supporting information, Fig.C1). High temporal resolution data for these stations is not freely available. Only 24 h accumulations were provided by AVAMET, which does not provide insight on the rainfall dynamics over time. Furthermore, AVAMET does not provide real-time access. For that reason, this data is not included in this study. If higher temporal resolution data is available in real-time, this would be interesting for future studies.

The high density of Netatmo PWSs in the Valencia region gives a unique opportunity to get insights in the rainfall dynamics and storm trajectory. Furthermore, PWS data are available in near real-time, with sometimes even a lower latency than gauges from official networks (Overeem et al., 2024). This gives the opportunity to use the data in operational settings, such as (flood) forecasting and improving decision making for disaster risk management. The network density of PWSs in the study area is even higher when other brands of PWSs are included. Combing dedicated rain gauge networks and PWSs reduces uncertainty in areal rainfall products. While rainfall observations from PWSs can contain incorrect data, this can be partially removed, reducing the uncertainty.

Another advantage of PWSs is that these instruments also measure additional meteorological variables in near real-time, including temperature, humidity, pressure and wind. These additional variables could be utilized in future studies to provide further insight into local meteorological conditions. However, further research is required to assess the reliability of these measurements, as wind measurements are generally more sensitive to setup and maintenance related errors than rainfall measurements.

#### 5.5 A posteriori reconstruction of the flood warning






Early warning systems are key in reducing the societal and economic impact of extreme rainfall and consequently flooding. Warnings over too large areas when only a part or none is affected by flooding, may erode public trust and eventually result in losing credibility of the warning system and lower preparedness among authorities or citizens to take necessary action. Therefore, flood early warning systems heavily rely on accurate forecasts with a high spatial and temporal resolution (Sawada et al., 2022).

Based on publicly available data, a timeline of the warnings and relevant processes was reconstructed (Fig. 10). Already six days in advance, on 23 October, a first message was issued by AEMET about the possible formation of DANA in the subsequent days (Maldita.es, 2024). The exact trajectory of the storm was difficult to forecast. The trajectory became more clear in the subsequent days, indicating that on 29 October most rainfall was expected, with more than 150 mm within 24 h. Special warnings were issued for amongst others the community of Valencia (area approximately 20,000 km<sup>2</sup>) and Murcia (area approximately 11,000 km<sup>2</sup>) (AEMET, 2024b, c, d, e). Between 06:30 and 08:00 AM on 29 October, multiple warnings

were issued by AEMET (Maldita.es, 2024). First, an orange warning was issued, indicating significant danger, which was raised within one hour to red, indicating extraordinary danger, anticipating more than 150 mm of rainfall. At that time, already 130 mm of rain has been observed by a PWS in the area. Around noon, water levels started to rise in Utiel. The Magro river started overflowing and a flood wave was propagating downstream. Around 15:00 PM the mayor of Utiel declared a state of emergency in Utiel. More than five hours after this, at 20:11 PM, an alert by a text message was issued to the inhabitants of the Province of Valencia by the National Emergency Center (Bubola, 2024). At that time, multiple PWSs have already measured 200 to 300 mm of rain over the past few hours (Fig. 5). Average rainfall in several catchments already exceeded 150 mm in the hours before the alert was issued (Figs. 6a and 7).

## 6 Conclusions







This study used low-cost rain gauges, referred to as personal weather stations (PWSs), to assess the torrential rainfall in Valencia, Spain, preceding and during 29 October 2024 floods. This rainfall event resulted in rainfall sums exceeding 300 mm within less than 24 h at several locations within this region. These daily sums are associated with return periods of 2,000 years or more. This event was particularly rare due to the widespread nature of the rainfall of more than 190 mm d<sup>-1</sup> over an area of 706 km<sup>2</sup>, with the catchment-average rainfall having an estimated return period exceeding 10,000 years. These return periods are uncertain due to the length of the timeseries, which contains less than 100 years of data. Additionally, they are sensitive to the method employed to fit the GEV distribution.

PWSs are owned and operated by citizens and therefore provide rainfall data independent from meteorological agencies. The data is available near real-time and the network density is larger compared to dedicated rain gauge networks. Data is highly correlated (0.94) and after applying a mean bias correction factor of 1.24, a bias of -4% is observed, indicating that rainfall estimates closely follow those from rain gauges operated by the Spanish Meteorological Agency (AEMET).

The analysis demonstrated the potential of using PWSs for monitoring rainfall, although it is limited to the areas with PWSs. The distribution of the PWSs made it possible to get insight in the rainfall dynamics in the Magro catchment, which was struck by torrential rainfall and consequently triggered floods. Already early in the morning intense rainfall occurred at several locations, exceeding 100 mm within a few hours. The estimated response time for this catchment is 2 h, suggesting that a flood was likely already triggered in the upstream parts of the catchment before the intense rainfall struck certain downstream locations in the afternoon. Multiple PWSs started having issues in recording rainfall between 15:10 and 16:15, indicating that subsequent data is uncertain and actual rainfall sums might have been higher at these locations. These timestamps can give an indication when flooding started and possibly affected power supply or even damaged PWSs located in the inundated area. Similar issues likely affected the AEMET stations, as there were 6 h data gaps.

Areal rainfall maps were produced using ordinary kriging. This resulted in catchment averages of more than 150 mm d<sup>-1</sup> across areas between 316 and 1661 km<sup>2</sup>. Differences between the quantitative precipitation estimates (QPE) from AEMET and PWSs were observed, likely arising from variations in the distribution and density between the rain gauge networks. Integrating the rainfall observations from both rain gauge networks would improve the spatial resolution and would therefore

reduce the uncertainty in QPE resulting from kriging. Other brands of PWSs exist, such as the PWSs collected by AVAMET or the platform from the Weather Observations Website (WOW) could be included as well to improve the coverage.

For these fast-responding catchments, warning systems heavily rely on accurate real-time rainfall observations with a high spatial and temporal resolution. As PWSs monitor rainfall in near real-time, flood forecasting systems could benefit from integrating PWS data with those from national meteorological and hydrological services. We recommend combining PWSs and dedicated rain gauge networks with weather radar data for real-time hydrological applications.

Data availability. Netatmo rainfall data can be accessed using an application programming interface (Netatmo, 2024, last access: 21 March 2025). Discharge data can be downloaded from the Confederación Hidrográfica del Júcar (https://saih.chj.es/chj/saih/glayer?t=a, last access: 21 March 2025) and from EStreams (Do Nascimento et al., 2024).

Appendix A

Appendix B

Appendix C


Author contributions. NR: conceptualization, data curation, formal analysis, investigation, methodology, software, validation, visualisation,
 writing. MH: funding acquisition, methodology, writing (review & editing). RU: conceptualization, funding acquisition, methodology, writing (review & editing).

Competing interests. One of the authors (MH) is member of the editorial board of HESS.

Acknowledgements. We would like to thank Adela Ramos Escudero and Patricia Mares Nasarre from the Faculty of Civil Engineering and Geosciences, Delft University of Technology, for sharing data and news articles relevant for this study. We thank Davide Wüthrich for discussions and for proofreading the manuscript. In addition, we would like to thank Jochen Seidel from the Institute for Modelling Hydraulic and Environmental Systems, University of Stuttgart, Germany and Georges Schutz from RTC4Water S.à r.l., Roeser, Luxembourg, for having insightful discussions about the Netatmo data as part of a short-term scientific mission funded by the OpenSense COST action (CA20136). This work is part of the Perspectief research programme "Future Flood Risk Management Technologies for rivers and coasts" with project number P21-23. This programme is financed by Domain Applied and Engineering Sciences of the Dutch Research Council (NWO).

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

**Figure 6.** a) Catchment-average rainfall over time (local time) based on PWS data for four catchments that were affected by torrential rainfall and or severely flooded, with the error bar indicating the uncertainty associated with kriging. Dashed line is the cumulative rainfall based on the hourly PWS data, dotted line is based on the 6-h AEMET data. Lower panel shows the catchment-average rainfall over the whole day based on b) AEMET stations c) PWSs and d) AEMET and PWSs combined. The numbers in the catchments indicate the average accumulated rain. Colours in (1) to (4) in a) correspond with the catchments indicated with the same colour in b), c) and d).

**Figure 7.** Sequence of maps of 6h catchment average rainfall (local time) on 29 October 2024, based on interpolating rainfall observations from both AEMET and PWSs gauges.

**Figure 8.** Return periods of annual daily maximum rainfall modelled using a GEV distribution, with  $\mu$  the location,  $\alpha$  the scale and  $\kappa$  the shape parameter. a) Estimated return periods for two rain gauges in the municipality of Valencia based on 87 and 58 years of data for Valencia and Valencia airport, respectively. b) Estimated return period of catchment-average daily rainfall for catchment ID2 in Fig. 1b based on 74 years of data. The arrow on the right indicates the estimated catchment-average rainfall on 29 October 2024 using AEMET rain gauge data. The red and blue shaded areas indicate the 95% confidence interval derived from bootstrapping by sampling with replacement.

**Figure 9.** Return periods of annual maximum discharges for three streamflow gauges (ID1-ID3 in Fig. 1b and Table 1), modelled using a GEV distribution, with  $\mu$  the location,  $\alpha$  the scale and  $\kappa$  the shape parameter. The red shaded area indicates the 95% confidence interval derived from bootstrapping by sampling with replacement. The blue square and star indicate the average daily discharge on 29 and 30 October, respectively.

**Figure 10.** A posteriori timeline reconstruction (hours in local time) of the warnings and relevant processes preceding and during the 29 October 2024 Valencia floods, based on Maldita.es (2024), Bubola (2024) and AEMET (2024b, c, d, e). Rainfall measured by PWSs corresponds to the PWSs indicated in Figs. 1c and 5.

**Figure A1.** a) Scatter plot of hourly rainfall between two pairs of PWSs (in Requena and Algemesí, indicated with B and E in Fig. 1c), with an inter-gauge distance of less than 1 km. The correlation coefficient (*r*) quantifies the relation between the stations.

Figure B1. Decorrelation distance of Mediterranean rainfall (d in km) versus time step (t in time), displayed on logarithmic axes. Fitted relation between range and time step are based on the relations reported by Lebel et al. (1987) and Berne et al. (2004).

**Figure C1.** Installation setup from the Davis PWS in Túris that recorded over 600 mm on 29 October 2024, obtained from: https://www.avamet.org/mx-fitxa.php?id=c20m248e02, last access: 28 November 2024. The PWS is indicated by the red circle.