# Peer review of "Torrential rainfall in Valencia, Spain, recorded by personal weather stations preceding and during the 29 October 2024 floods"

_EGUsphere, 2025_

## Referee Comment (RC2)

Torrential rainfall in Valencia (Spain) recorded by personal weather stations preceding and during the 29 October 2024 floods

By Nathalie Rombeek et al

Sep 11 2025

Rev. B

Decision: Major revisions

General

This manuscript focuses on" quantify and describe the spatial and temporal structure of the rainfall event occurred on 29 October 2024 exceeding 300 mm within less than 24 h, that caused devastating floods in the province of Valencia in Spain. Using rainfall observations from approximately 225 personal weather stations (PWSs), low-cost commercial devices primarily operated by citizens. The network density of PWSs is ~7 times higher compared to the dedicated rain gauge network operated by the Spanish Meteorological Agency (AEMET) in the province of Valencia, allowing a more detailed analysis of the spatial and temporal rainfall dynamics. Overall, interesting study using low cost sensors' measurements to quantify the precip/flooding conditions. Although, scientifically interesting work, there are several limitations in the analysis and needs to be addressed before going to be published.

Major issues:

1. Needs to show precip rate measurements versus a pluvio/distrometer based measurements. Tipping Buckets have their own issues as you described here.
2. Please provide a comparison of two different data set obtained from different two sensors.
3. What are the averages of the observations used? Can you compare the 5 min measurements?
4. Ln 25; what is high level? 500 mb or 200 mb etc.
5. Fig. 3; what the blue circles, how averages are made? >30 mm/hr there are large errors at about 20 mm/hr. Do you have pluvio measurements? 94% relationship? Provide absolute difference between two sets using PR>30 mmhr-1
6. Fig 4; Better to show dPR between 2 measurements and plot against PR measurements.

7. Fig. 4; PR>50 mm/hr, PWS2 has values at about 20-25 mm/hr less than PWS1. 25-30% diffs.
8. Fig. 5c; second event value cant be located at the PA (accum precip); why is that?
9. Data Uncertainty; you need an independent instrument for this analysis; like pluvio or Distrometer.
10. Now, you have wind measurements, did you look at wind speed and directional effect on precip rate and amount?
11. In the calculations for flood volume, did you look at the mountain slope effect? Rather than only precip impact on flooding?

Overall, based on above issues and clarifications, this works needs major issues.

---

## Author Response (AR1)

**Reviewer 1**

Overall, a well written paper that makes a clear case for the value of high spatial resolution rain fall measurements from personal weather stations. The authors have made well justified decisions in their data analysis.

We would like to thank the referee for the constructive feedback on our manuscript. We highly appreciate the time and effort to read our manuscript and provide us with new insights on our research. We took the comments into account when revising the manuscript.

We separated the different comments, shown in *italic*, from our replies (in regular font) below. In **bold** we provide our revised text.

**Major comment:**

The spatial sequence of rainfall during the flooding event is not clearly shown with the existing Figures. It is hard for the reader to relate the time series shown in Fig 5 and Fig 6a to the big picture of how the event unfolded. Suggest add a multi-panel plot that shows the sequence of maps of average rainfall in each catchment for shorter time intervals than the storm total ones shown in Figure 6b-d. Perhaps 3 or 4 hour time intervals and use the PWS and AEMET combined data. If use 4 hours as time interval, it would be 4 subplots maps, hours 4-7, hours 8-11, hours 12-15, hours 16-19 would cover nearly all of the period of interest. Discussion of this new figure would complement discussion of the warning timeline in Section 5.5.

We thank the reviewer for this suggestion. We recognize that the temporal evolution displayed in Figs. 5 and 6a cannot easily be translated into the corresponding location. For that reason, we adjusted Figs. 5 and 6a. In addition, we included a new figure to help to understand the spatio-temporal evolution of the storm in the geographical context and in more detail and we added a paragraph discussing this new figure. We only have access to 6-h AEMET rainfall data. For that reason, we created a sequence of maps of 6h catchment average rainfall. In Fig. 5, we included the outline of the Magro catchment to each panel and marked the corresponding PWS location for the timeseries shown:

Figure 5. Time series of the 5-min rainfall observations from six selected PWSs that recorded more than 100 mm d- 1 on 29 October 2024 in local time. Letters in the figure correspond with the locations (indicated with red capital letters) in Fig. 1c. Left y-axis shows the intensity in mm  $h^{-1}$ , right y-axis the cumulative rainfall in mm. The red dashed line indicates when measurements are (temporarily) interrupted and total rainfall sums become uncertain. In the black dotted box the outline of the Magro catchment is shown, including the location of the PWS displayed in the panel (red dot) and the locations of the other five PWSs (black dots).

In Fig. 6a we also added the outline of the specific catchment next to the panels. On the right side of the panels (1-4) the four catchment outlines are shown, with the coloured one corresponding to the timeseries displayed in that row.

Figure 6. a) Catchment-average rainfall over time (local time) based on PWS data for four catchments that were affected by torrential rainfall and or severely flooded, with the error bar indicating the uncertainty associated with kriging. Dashed line is the cumulative rainfall based on the hourly PWS data, dotted line connects the 6-h AEMET data points. Lower panel shows the catchment-average rainfall over the whole day based on b) AEMET stations c) PWSs and d) AEMET and PWSs combined. The numbers in the catchments indicate the average accumulated rain. Colours in (1) to (4) in a) correspond with the catchments indicated with the same colour in b), c) and d).

Figure 7 shows the spatio-temporal evolution of the storm, using 6-h AEMET and PWS data:

Figure 7 Sequence of maps of 6h catchment average rainfall (local time) on 29 October 2024, based on interpolating rainfall observations from both AEMET and PWSs gauges.

We discussed Fig. 7 in Section 4.4 Spatio-temporal evolution, lines 273-281:

"The spatio-temporal evolution of the storm in a geographical context is shown in Fig. 7. These maps show that during the first 6 hours (00:00 - 06:00, local time) the rainfall was concentrated over catchments located in the south of the study area. Over the next 6 hours, the storm's spatial extent increased, predominantly growing in the north-westerly direction. Rainfall intensities increased substantially, with the Magro catchment and adjacent catchments receiving the most rainfall. Between 12:00 and 18:00, the spatial extent of the storm decreases, with highest rainfall intensities still occurring in the central part of the study area. Based on the available rainfall data, the storm's spatial extent and rainfall intensities appear to decrease further during the last 6 hours, with most rainfall concentrated in the catchments in the central and northern parts of the study area. Note that the rainfall observations during the end of this period are uncertain due to for example (temporary) power failure (see Fig. 5)."

**Minor comments:**

- Suggest change title to: "Torrential rainfall in Valencia, Spain recorded by personal weather stations preceding and during the 24 October 2024 floods" (i.e. remove parentheses)
  - We changed the title into: "Torrential rainfall in Valencia, Spain, recorded by personal weather stations preceding and during the 29 October 2024 floods"
- Overuse and misuse of conjunctions like "However" and "Nevertheless". For example, line 181: "However, the timing of the peaks and the rainfall depth varied within this region." Better would be "The timing of the peaks and the rainfall depth varied within this region." since authors are not contradicting previous sentence.
  - You have a point. We changed L188 into: "The timing of the peaks and the rainfall depth varied within this region."
  - Note that we will go through the manuscript and correct the sentences where we overused or misused such conjunctions.
- Line 42: Suggest add a sentence as part of this paragraph. Especially in mountainous areas, radar beams can be blocked over areas of interest.
  - Good point. We will add a sentence in the introduction L42-44 about the challenges of

weather radars in mountainous areas, such as beam blocking. "Moreover, rainfall estimation in mountainous areas is particularly challenging for weather radars, as mountain ranges can (partially) block radar beams (Pellarin et al., 2002 and Germann et al., 2006)."

- Caption for Table 1: For clarity please change "Mean discharge and peak discharge for gauge ID1, ID2, ID3..." to "Mean discharge and mean yearly peak discharge for gauges ID1, ID2, ID3..."
  - Thanks. We changed the caption for Table 1 into: "Mean discharge and mean yearly peak discharge for gauges ID1, ID2, ID3 are calculated from mean daily streamflow data in Do Nascimento et al. (2024)."
- Line 140, if this is the correct interpretation, suggest change "the nugget is assumed to be negligible" to "the nugget is assumed to be zero"
  - You are right. We changed L147-148 from "Similarly to Van de Beek et al. (2011), the nugget was assumed negligible." into "Similarly to Van de Beek et al. (2011), the nugget was assumed to be zero."
- Line 215, please cite reference for Hortonian overland flow
  OK, we added the reference for Hortonian overland flow reference to L221-223:
  "These rainfall intensities likely exceeded soil infiltration capacities, generating
  Hortonian overland flow (Horton, 1933) across the hilly catchment and potentially
  triggering a first flood wave."
- Line 324-328, for clarity suggest revise "However, selecting smaller catchment areas would increase the uncertainty in the interpolated rainfall estimates based on point measurements, particularly for areas with no or only a few rain gauges. Rainfall estimates from weather radars can mitigate these gaps. Additionally, these delinations do not necessarily match the upstream areas of streamflow gauges. Nevertheless, given the unavailability of of high temporal resolution discharge time series, this is not a limitation for this study." To "Selecting smaller catchment areas would increase the uncertainty in the interpolated rainfall estimates based on point measurements, particularly for areas with no or only a few rain gauges. Additionally, smaller catchment areas would not necessarily match the upstream areas of streamflow gauges." [not clear how the Nevertheless sentence relates to the rest of the paragraph so suggest taking it out].

We would like to thank the reviewer for pointing out that this paragraph was not clear and taking the time to provide a suggestion to improve the clarity.

Our intention was to explain that the relatively large catchment areas used in this study are not necessarily a limitation in the context of catchment average rainfall and in a hydrological context.

We took the suggestion of the reviewer into account and changed it into (see lines 340-344): "Selecting smaller catchment areas would increase the uncertainty in the interpolated rainfall estimates based on point measurements, particularly for areas with no or only a few rain gauges. Rainfall estimates from weather radars can potentially mitigate this gap. From a hydrological perspective, using relatively large catchment areas is not a limitation either, especially given the unavailability of high temporal resolution discharge observations. Additionally, smaller catchment areas would not necessarily match the upstream areas of streamflow gauges."

• Line 337. As a reader I got confused by this paragraph:

"The Poyo catchment, is characterized by an ephemeral stream, as it primarily depends on rainfall (Camarasa-Belmonte, 2016). This streamflow-gauge is not included in the EStreams database employed in this study. However, the mean and maximum specific peak flow for 37 flash flood events in this catchment, occurring between 1989 and 2007, were 16 and 246 mmd–1, respectively (Camarasa-Belmonte, 2016). Compared to this average peak flow, the recorded discharge on the 29th and 30th with 147 and 433 mm d–1 is approximately 9 and 27 times higher, respectively. The average discharge on the 30th with 433 mm d–1 is significantly higher (1.75 times) compared to the maximum peak flow in this period. However, due to the short duration of the historical record used in Camarasa-Belmonte (2016) (18-years), no definitive conclusions can be drawn regarding the rarity of this event."

**Some points of confusion:**

"This streamflow-gauge is not included in the EStreams database employed in this study." Since streamflow-gauge is included in Table 1 as ID4, I think you mean in terms of computing mean and mean yearly peak.

It is not clear what numbers are being referred to and where it is coming from for the "average discharge" and "maximum peak flow in this period".

Suggest revise to read: "The Poyo catchment, is characterized by an ephemeral stream, as it primarily depends on rainfall (Camarasa-Belmonte, 2016). For 37 flash flood events in this catchment occurring between 1989 and 2007, the mean was 16 mm d-1 and maximum specific peak flow was 246 mm d-1 (Camarasa-Belmonte, 2016). The daily recorded discharges from the Poyo catchment on the October 29th and 30th of 147 and 433 mm d-1 are approximately 9 and 27 times higher, respectively (Table 1). Due to the short duration of the historical record used in Camarasa-Belmonte (2016) (18-years), no definitive conclusions can be drawn regarding the rarity of this event."

We used the EStreams database to access discharge timeseries and put the 29 October 2024 event into perspective. However, ID4 is not included in the EStreams database, so mean and mean yearly peak discharge could not be derived from this gauge. The discharge prior, during and after the 29 October 2024 flood event was available at the local waterboard (Confederacion Hidrografica del Jucar). The waterboard provides only access to data for the most recent three months. Consequently, neither the mean nor the mean yearly peak discharge could be derived from timeseries provided by the local waterboard.

To put the 29 October 2024 discharge levels at ID4 into perspective, we reviewed existing literature that studied the same catchment. This is where the numbers came from. We understand that this is unclear from the paragraph. Therefore we revised the paragraph to improve the readability. In addition, we noticed a small typo: instead of 37 flash floods it should have been 38 flash floods between 1989 and 2007 in the Poyo catchment.

**We revised this paragraph from:**

"The Poyo catchment, is characterized by an ephemeral stream, as it primarily depends on rainfall (Camarasa-Belmonte, 2016). This streamflow-gauge is not included in the EStreams database employed in this study. However, the mean and

maximum specific peak flow for 37 flash flood events in this catchment, occurring between 1989 and 2007, were 16 and 246 mm d-1, respectively (Camarasa-Belmonte, 2016). Compared to this average peak flow, the recorded discharge on the 29th and 30th with 147 and 433 mm d-1 is approximately 9 and 27 times higher, respectively. The average discharge on the 30th with 433 mm d-1, is significantly higher (1.75 times) compared to the maximum peak flow in this period. However, due to the short duration of the historical record used in Camarasa-Belmonte (2016) (18-years), no definitive conclusions can be drawn regarding the rarity of this event.

Into (lines 352-365): "The Poyo catchment is characterized by an ephemeral stream (Camarasa-Belmonte, 2016). No historic discharge timeseries longer than three months were available for the streamflow gauge in this catchment (ID4). Consequently, mean and mean yearly peak discharge could not be derived for this gauge. To put the 29 October 2024 discharge levels at ID4 into perspective, we reviewed literature that studied the same catchment. Camarasa-Belmonte (2016) collected discharge data from 38 flash flood events that occurred in the Poyo catchment between 1989 and 2007. They found that the mean and maximum specific peak flow from these 38 events were 16 and 246 mm d-1, respectively. In comparison, the recorded discharge on 29 and 30 October 2024 were 147 and 433 mm d-1 (Table 1), approximately 9 and 27 times higher than the mean peak flow from the 38 flash flood events recorded between 1989 and 2007. The average discharge on 30 October 2024 was significantly higher (1.75 times) than the maximum specific peak flow recorded between 1989 and 2007. Due to the short duration of the historical record used in Camarasa-Belmonte (2016) (18-years), no definitive conclusions can be drawn regarding the return period of this event."

**References:**

Pellarin, T., Delrieu, G., Saulnier, H., Andrieu, B., Vignal, and Creutin, J.: Hydrologic Visibility of Weather Radar Systems Operating in Mountainous Regions: Case Study for the Ardèche Catchment (France). J. Hydrometeor., 3, 539–555, https://doi.org/10.1175/1525-7541(2002)003%3C0539:HVOWRS%3E2.0.CO;2, 2002.

Germann, U., Galli, G., Boscacci, M. and Bolliger, M.: Radar precipitation measurement in a mountainous region. Q.J.R. Meteorol. Soc., 132: 1669-1692, <a href="https://doi-org.tudelft.idm.oclc.org/10.1256/qj.05.190">https://doi-org.tudelft.idm.oclc.org/10.1256/qj.05.190</a>, 2006.

Horton, R. E.: The role of infiltration in the hydrologic cycle, EOS Trans. Am. Geophys. Union, 14, 446–460, 10.1029/TR014i001p00446, 1933.

**Reviewer 2**

Torrential rainfall in Valencia (Spain) recorded by personal weather stations preceding and during the 29 October 2024 floods

By Nathalie Rombeek et al

Sep 11 2025

Rev. B

Decision: Major revisions

**General**

This manuscript focuses on" quantify and describe the spatial and temporal structure of the rainfall event occurred on 29 October 2024 exceeding 300 mm within less than 24 h, that caused devastating floods in the province of Valencia in Spain. Using rainfall observations from approximately 225 personal weat her stations (PWSs), low-cost commercial devices primarily operated by citizens. The network density of PWSs is ~7 times higher compared to the dedicated rain gauge network operated by the Spanish Meteorological Agency (AEMET) in the province of Valencia, allowing a more detailed analysis of the spatial and temporal rainfall dynamics. Overall, interesting study using low cost sensors' measurements to quantify the precip/flooding conditions. Although, scientifically interesting work, there are several limitations in the analysis and needs to be addressed before going to be published.

We would like to thank the referee for the constructive feedback on our manuscript. We highly appreciate the time and effort to read our manuscript. We took the comments into account when revising the manuscript.

We separated the different comments, shown in *italic*, from our replies (in regular font) below. In **bold** we provide our revised text.

**Major issues:**

1. Needs to show precip rate measurements versus a pluvio/distrometer based measurements. Tipping Buckets have their own issues as you described here.

We agree with the reviewer that it is important to make a comparison with accurate rain gauges (such as pluvio or disdrometers) in order to quantify the performance of tipping bucket gauges from PWSs. Previous studies already performed such comparisons. For example, de Vos et al. (2017) used an experimental setup to show that under ideal circumstances (i.e. installed and maintained according to World Meteorological Organization standards), three PWSs recorded rainfall with high accuracy. The reference station was a floating-type-gauge from Royal Netherlands Meteorological Institute (KNMI), which estimates cumulative rainfall every 12 s by measuring the displacement of a float placed in a reservoir. More recently, Rombeek et al. (2025) performed a systematic long-term analysis involving PWS rainfall observations and focused on the highest rainfall events over different accumulation intervals and seasons over a 6-year period. That study compared rainfall observations from PWSs against KNMI's

professional rain gauge network (float-type), and showed an overall high performance of the PWSs, especially for longer accumulation intervals.

Building on these previous findings we did not re-address this comparison. As it was not clear from the manuscript why we did not perform this comparison, we added the following text in the introduction (see lines 59-64), to make it clear for the reader that previous studies already investigated the performance of PWSs compared to float-type-gauges:

"For example, De Vos et al. (2017) used an experimental setup to show that under ideal conditions (i.e. installed and maintained according to World Meteorological Organization standards), three PWSs collocated with a floating-type gauge from the Royal Netherlands Meteorological Institute (KNMI) recorded rainfall with high accuracy. More recently, Rombeek et al. (2025) performed a systematic long-term analysis of PWS rainfall observations by comparing them against KNMI's professional rain gauge network (float-type), and reported an overall high performance of the PWSs, especially for longer accumulation intervals."

Since rainfall observations from PWSs are prone to PWS-related-errors, such as those related to inappropriate setups and a lack of maintenance, we found it essential to compare them with the dedicated rain gauge network from AEMET (which are presumably installed and maintained according the WMO guidelines), to assess their reliability for this specific event. This comparison, already presented in the manuscript in Section 4.1 (Fig. 3 and lines 163-173), was used to assess the accuracy of the rainfall observations from the PWSs against the dedicated rain gauge network from AEMET. To limit spatial representativeness errors in this comparison, we selected for each AEMET rain gauge the nearest PWS, with a maximum inter-gauge distance of 10km. If the inter-gauge distance exceeded 10 km, the AEMET-PWS pair was discarded. This resulted in 24 AEMET-PWS pairs that we used for the comparison. This comparison shows that there is a high correlation coefficient (r = 0.94) between the datasets and confirms that there is a small underestimation (bias = -0.04, see Fig. 3). Therefore, we consider the rainfall observations from the PWSs in the study area sufficiently reliable for the purpose of this study.

2. Please provide a comparison of two different data set obtained from different two sensors. In Figure 3 we compared two different datasets from two different rainfall sensors, namely rainfall observations from PWSs with the dedicated rain gauge network from AEMET, see also answer to point 1.

**3. What are the averages of the observations used? Can you compare the 5 min measurements?**

The rainfall observations from PWSs have a temporal resolution of 5 min. Rainfall observations from AEMET were only freely available with a 6h temporal resolution, unfortunately. We agree that a 5 min comparison between the rain observations from these different rain gauge networks would have been preferable. However, to the best of our knowledge, a higher temporal resolution from AEMET gauges is not freely available.

**4. Ln 25; what is high level? 500 mb or 200 mb etc.**

The weather system that caused this flood is a cut-ff low, referred to as DANA is Spanish (Isolation Depression at high levels), see also Faranda et al. 2024. In the initial phase and maturity, DANAs typically show up on the high-altitude maps (at 250,

**5. Fig. 3; what the blue circles, how averages are made? >30 mm/hr there are large errors at about 20 mm/hr. Do you have pluvio measurements? 94% relationship? Provide absolute difference between two sets using PR>30 mmhr-1**

In figure 3 we compared the daily rainfall observations (mm/d) of the dedicated rain gauge network from AEMET with the daily rainfall observations from personal weather stations. Each blue circle is representing another AEMET-PWS pair, with the rainfall observations from the AEMET stations on the x-axis and from the PWSs on the y-axis. The statistics shown in Fig. 3 are averaged over the 24 AEMET-PWS pairs. The 0.94 indicates the correlation coefficient (see also caption of Fig. 3). For more details, we refer to our answer to point 1.

**6. Fig 4; Better to show dPR between 2 measurements and plot against PR measurements.**

We believe that a scatter plot (indicating random error) and double mass plot (indicating bias error) already provide a complete picture of the correspondence to the nearby PWS. The figures already show graphically the effect of rainfall intensity on the differences between the nearby PWSs at a 5-min temporal scale (see Appendix Fig.A1 for the hourly scatterplot). We decided not to include a regression analysis in Fig. 4a, because both datasets come from PWSs and none of them can be considered a reference. To study the effect on potential biases we included the double mass analysis, see Fig. 4b.

**7. Fig. 4; PR>50 mm/hr, PWS2 has values at about 20-25 mm/hr less than PWS1. 25- 30% diffs.**

We agree with the reviewer that there are notable differences between the PWSs. In lines 54-57 we mentioned that rainfall observations from PWSs are not perfect, but prone to several sources of error. These PWS-related-errors, such as sub-optimal installation and maintenance, and the fact that the PWSs are not collocated, likely play a role in explaining the differences shown in Fig. 4.

**8. Fig. 5c; second event value cant be located at the PA (accum precip); why is that?**

The accumulated precipitation is included in Fig. 5c. The second event is also visible in the dashed line (indicating a rainfall sum of around 200 mm before the second peak and approximately 300mm after the event). We also checked all other panels in Fig. 5, which display the precipitation accumulated over time. We were not able to find one instance where the accumulated precipitation is not showing the second event.

**9. Data Uncertainty; you need an independent instrument for this analysis; like pluvio or Distrometer.**

We agree with the reviewer that an independent instrument is essential for assessing data uncertainty. In the manuscript we already compared PWSs in the Valencia region with the dedicated rain gauge network from AEMET, see also our answer to point 1.

**10. Now, you have wind measurements, did you look at wind speed and directional effect on precip rate and amount?**

We recognize that there can be a significant wind effect in rainfall measurements and that this could be interesting for future studies. However, in our study we did not take wind measurements into account, as PWSs are not necessarily equipped with a wind module. Since these wind modules are an optional extension that needs to be bought by citizens, only a limited number of PWSs provide wind in addition to rainfall data (e.g. less than 17% of the PWSs in the Netherlands have a wind module). These wind measurements are also more sensitive to setup and maintenance related errors compared to the rainfall observations from PWSs. Further research is required to assess the reliability of these measurements, which lies outside the scope of this study. We added a sentence to the manuscript mentioning this as a potential subject for future research (see lines 389-391):

"However, further research is required to assess the reliability of these measurements, as wind measurements are generally more sensitive to setup and maintenance related errors than rainfall measurements."

11. In the calculations for flood volume, did you look at the mountain slope effect? Rather than only precip impact on flooding?

The streamflow observations we presented in our study were obtained from the local waterboard (Confederacion Hidrografica del Jucar). These values represent the observed flows on the days before, during and after the flood event.

We agree with the reviewer that catchment properties, including slope and land use, play an important role in the hydrological response and should not be overlooked. The current study focuses specifically on describing and quantifying the spatial and temporal structure of the flood-producing rainfall event on 29 October 2024. For this reason, we did not look at catchment properties, such as slope and land use, which likely affected the hydrological response.

Overall, based on above issues and clarifications, this works needs major issues

**References:**

de Vos, L., Leijnse, H., Overeem, A., and Uijlenhoet, R.: The potential of urban rainfall monitoring with crowdsourced automatic weather stations in Amsterdam, Hydrol. Earth Syst. Sci., 21, 765–777, https://doi.org/10.5194/hess-21-765-2017, 2017.

Rombeek, N., Hrachowitz, M., Droste, A., and Uijlenhoet, R.: Evaluation of high-intensity rainfall observations from personal weather stations in the Netherlands, Hydrol. Earth Syst. Sci., 29, 4585–4606, https://doi.org/10.5194/hess-29-4585-2025, 2025.

Llasat, MC., Martín, F. & Barrera, A. From the concept of "Kaltlufttropfen" (cold air pool) to the cut-off low. The case of September 1971 in Spain as an example of their role in heavy rainfalls. *Meteorol. Atmos. Phys.* **96**, 43–60, <a href="https://doi.org/10.1007/s00703-006-0220-9">https://doi.org/10.1007/s00703-006-0220-9</a>, 2007.

Faranda, D., Alvarez-Castro, MC., Ginesta, M., Coppola, E., Pons, FME.: Heavy precipitations in October 2024 South-Eastern Spain DANA mostly strengthened by human-driven climate change, ClimaMeter, Institut Pierre Simon Laplace, CNRS, <a href="https://doi.org/10.5281/zenodo.14052042">https://doi.org/10.5281/zenodo.14052042</a>, 2024.